# The Potential Contribution of Caveolin 1 to HIV Latent Infection

**DOI:** 10.3390/pathogens9110896

**Published:** 2020-10-27

**Authors:** Bikash Sahay, Ayalew Mergia

**Affiliations:** Department of Infectious Disease and Immunology, University of Florida, Gainesville, FL 32611, USA; sahayb@ufl.edu

**Keywords:** HIV, caveolin 1, latent infection

## Abstract

Combinatorial antiretroviral therapy (cART) suppresses HIV replication to undetectable levels and has been effective in prolonging the lives of HIV infected individuals. However, cART is not capable of eradicating HIV from infected individuals mainly due to HIV’s persistence in small reservoirs of latently infected resting cells. Latent infection occurs when the HIV-1 provirus becomes transcriptionally inactive and several mechanisms that contribute to the silencing of HIV transcription have been described. Despite these advances, latent infection remains a major hurdle to cure HIV infected individuals. Therefore, there is a need for more understanding of novel mechanisms that are associated with latent infection to purge HIV from infected individuals thoroughly. Caveolin 1(Cav-1) is a multifaceted functional protein expressed in many cell types. The expression of Cav-1 in lymphocytes has been controversial. Recent evidence, however, convincingly established the expression of Cav-1 in lymphocytes. In lieu of this finding, the current review examines the potential role of Cav-1 in HIV latent infection and provides a perspective that helps uncover new insights to understand HIV latent infection.

## 1. Introduction

The development of HIV vaccines with adequate protection remains elusive. However, there are encouraging studies that will help promote HIV vaccine development. The phase 2b clinical trial in Thailand RV144 showed a reduced acquisition of HIV infection with a 31% efficacy [1]. Although these results are significant and raised hopes, we have not yet reached the goal of attaining an effective HIV vaccine. Combinatorial antiretroviral therapy (cART) suppresses HIV replication to undetectable levels and has been effective in prolonging the lives of HIV infected individuals. The suppression of viral replication achieved by many patients on antiretroviral therapy (ART) initially raised hopes for virus eradication. It is now well documented that HIV persists in a small reservoir of latently infected resting CD4+ cells of patients on ART [2,3,4,5,6,7]. HIV establishes latent infection in the blood and lymphoid tissues as well as multipotent hematopoietic progenitor cells in the bone marrow [8,9]. Besides, monocytes, dendritic cells, and macrophages can contribute to viral persistence as latently infected cells [10,11,12,13,14,15]. The central nervous system (CNS) is another important target for HIV infection [16,17,18]. Many drugs are restricted from crossing the blood–brain and blood–CSF barriers and are therefore suboptimal in penetrating the CNS. Consequently, the CNS serves as a viral reservoir that remains untouched to cART therapy. Residual viral reservoirs in the gastrointestinal (GI) and genitourinary tract, as well as lymphoid tissues in patients under drug treatment, can be an additional obstacle for curing HIV [7,19,20,21,22,23,24]. Despite the advancement of combinatorial drug therapies that have aided in treating HIV infected patients and helped prolong life successfully, current drug therapy is not curative due to HIV’s ability to establish a latent infection that persists in reservoir cells. Consequently, HIV infected individuals will remain on cART treatment throughout their lives. Thus, the use of cART has converted HIV infection into a manageable chronic disease. Although a manageable disease HIV infected individuals under cART are prone to other Non-AIDS diseases [25,26,27]. This enforces the need for a complete understanding of HIV latent infection and subsequently find a strategy for eradication and cure.

Latent infection occurs when the HIV-1 provirus becomes transcriptionally inactive, and several contributing mechanisms already identified as repressive chromatin structure, transcriptional interference from adjacent transcriptional units, increase in histone deacetylase (HDAC) and methyltransferase (HMT) activity and DNA methylation of proviral DNA [28,29,30,31,32]. Reactivation of HIV latent infected cells in the presence of cART would ideally allow for recognition of latent infected cells by the immune system and consequently clearance and cure. Several strategies have been described and examined to reverse latent infection to target and eradicate HIV [28,29,30,31,32,33]. A latent reversing agent (LRA) is one approach as a “shock and kill” strategy in combination with cART to achieve functional cure HIV infection [33,34,35,36,37]. Examples of LRAs tested include protein kinase C (PKC) agonists (e.g., PMA, prostratin, bryostatin, and ingenol molecules) [33,36,38,39,40,41,42,43] and histone deacetylase inhibitors (HDACi; e.g., vorinostat/SAHA, panobinostat, and romidepsin) [33,34,35,44,45,46,47,48,49,50,51], indirect activator of Akt signaling pathways (Disulfiram) [33,52,53,54,55], and CCR5 antagonist (maraviroc) [56]. A recent promising study of reactivation virus in latently CD4+ T cells with the HDACs inhibitor vorinostat shows the cells were susceptible to HIV antigen-specific CD8+ T cells mediated killing in vitro [34]. However, further evidence suggests this process alone may not be sufficient. Global activation of T cells to reactivate virus to target for therapeutic has resulted in a toxic level of immune activation [28,30,32,57,58]. A variety of molecules that induce HIV transcription or inhibit HDACs or HMTs also suffer from lack of specificity, and their safe and efficient in vivo application and clinical trials have yet to be established [28,30,32,34,44,59,60,61,62]. Furthermore, LRAs are inefficient and have limitations in disrupting latency in other cell types such as macrophages and microglial cells. Alternative approaches are emerging to tackle HIV latent infection based on HIV latent promoting agents (LPAs) and proviral DNA scission. The LPA approach is described as “block and lock,” and the principle behind is to effectively silence (block) the promoter region and restrict (lock) viral transcription subsequently impairing viral rebound. For the induction of latency, several methods have been used; however, all these efforts can be categorized in three groups. First approach was to use small inhibitory RNA (siRNA) or natural long non-coding RNA (lncRNA) to prevent either the transcription of HIV-1 RNA or initiate a repressive chromatin modification at the proviral DNA. The most common among these approaches were, siPromA, LTR363, si2A, and S4. All these targeted NF-κB binding region of the proviral DNA to modify the chromatin structure to make them inaccessible for transcription. The second approach was to prevent the Tat, the potent transactivator of HIV-1 transcription. The third approach was used to modulate the signaling by small molecule inhibitors of mTOR pathway. Recently, the use of, the clustered regularly interspaced short palindromic repeats (CRISPR), which rely on CRISPR RNAs (crRNAs) to direct cleavage of complementary sequences via the nuclease activity of CRISPR-associated (Cas9) protein system, has exploded and become a great tool to knock out genes. HIV provirus deletion has been accomplished in vitro using the CRISPR/Cas9 system [63,64,65]. Although targeting proviral genome using CRISPR/Cas9 provided initial success, general variability in viral sequences and escape from the CRISPR by mutation posed hurdles in successful use of the technology [66,67]. Despite these advances, latent infection remains a major hurdle to cure HIV infected individuals. Therefore, to accomplish a complete purge of latent reservoir, further research is critical to understand better the factors that influence and maintain HIV latency. Major critical factors that are missing include complete understanding of the mechanism HIV latency, cell membrane cues that sends the signal for the establishment of latent infection, and markers that help identify latently infected cells.

## 2. Caveolin-1

One molecule that can be important in HIV latent infection is Caveolin 1 (Cav-1). Contrary to the previous belief, recent findings show that human T cells can express Cav-1 [68,69,70]. Along with this finding, the multiple functions of Cav-1 with relevance to cell regulation can have important implications in HIV latent infection. Caveolin 1 (Cav-1), a 21–24-kDa scaffolding protein, is an important structural component of the caveolae organelle [71]. Cav-1 is also important in establishing specific lipid microdomains (non-caveolar caveolin lipid raft (NCCLR)) and helps compartmentalize signal pathways. Functional studies have shown that Cav-1 is involved in a wide range of cellular processes (Table 1), including cell cycle regulation, signal transduction, endocytosis, cholesterol trafficking, and efflux [72,73,74,75,76,77,78]. Furthermore, Cav-1 engages in crosstalk with the actin cytoskeleton and contributes to mechanosensing and adaptation to various mechanical stimuli and environmental changes that include microbe infection [70,79,80,81,82,83,84,85,86]. Therefore, Cav-1 regulates multiple signaling cascades as well as provides crosstalk with different molecules, and altered expression or/and any perturbation of Cav-1 positioning in the vicinity of the plasma membrane can affect signaling and crosstalk of different biological molecules. Cav-1 is highly expressed in terminally differentiated or quiescent cells, including muscle cells, adipocytes, endothelial cells, monocytes, macrophages, dendritic cells, microglia, and astrocytes [74,87,88,89,90,91,92,93,94,95,96,97]. Initially, it was thought that lymphocytes do not express Cav-1 [74,87,88,89,90,91,92,93,94,95,96,97]. However, several recent reports reveal both T and B primary lymphocytes express Cav-1 at low levels [68,69,70,83,98,99]. The expression of Cav-1 protein at times can be difficult to detect by standard methods. Further studies reveal that Caveolin-1 is important in B cell antigen receptor (BCR) and T cell antigen receptor (TCR) basal membrane organization and also reorganization upon stimulation. Several of the proteins that interact with Cav-1 are suggested to be involved in TCR-regulated membrane dynamics and intracellular signaling [68,69,83,100,101,102]. Cholesterol plays an important role in the resting or activation TCR either by stabilizing the resting TCR conformation or by increasing sensitivity to antigen and cooperation, respectively [83,103,104,105,106,107,108,109,110,111,112]. Since Cav-1 is engaged in cholesterol trafficking and efflux, it can regulate the conformational stage of TCR or other lipid rafts. In addition, Cav-1 is an essential component of the lipid raft platform for the recruitment of signaling proteins to the plasma membrane. Signal transduction most probably happens by linking the plasma membrane and the actin cytoskeleton [83,113]. Therefore, regulation of TCR and other lipid rafts integrates environmental cues such as the concentration of ligand or cholesterol metabolism to modulate TCR responses and T cell resting/activation. NCCR can, thus, regulate membrane dynamics compartmentalization upon receptors activation.

## 3. Potential of Cav-1 in Keeping the Chromatin Open

Cav-1 maintains its presence at the cell surface and segregates different signaling components for the ligand-dependent signaling. In diabetes and ozone injury [116] models, Cav-1 negatively regulates PI3K signaling, a signaling that prevents latency by the activation of NF-κB p65. The expression of Cav-1 itself is prevented by the activation of NF-κB [117]. In macrophages, Cav-1 association with TLR4 prevents pro-inflammatory cytokine production [114] that could add to the viral replication by further activation of NF-κB. Additionally, TLR4 signaling triggers a cascade to degrade Cav-1 by a ring-type E3 ubiquitin ligase, zinc, and ring finger 1 (ZNRF1) [118]. Cav-1 deficient mice maintain a constant inflammatory state; however, T cells deficient in Cav-1 tend to develop into non-inflammatory regulatory T cells. Further research is needed to reconcile these counterintuitive findings. In several human inflammatory diseases, Cav-1 was found lower than the normal levels, suggesting a strong anti-inflammatory role of Cav-1. The cytokine and TLR signaling simultaneously trigger cascade for Mitogen Associated Protein Kinases that control chromatin structure by various mechanisms. Based upon these correlations, we can predict Cav-1 can prevent the transcription of HIV-1 proviral DNA at different levels. Since Cav-1 also ushers signaling components to lipid raft, a careful assessment of Cav-1 in different cells and different condition is necessary (Figure 1).

## 4. Caveolin-1 and the Potential Link to HIV Latent Infection

The participation of Cav-1 in environmental cues makes this molecule an important regulator of cell physiology. Interestingly, activation or antigenic stimulation of the TCR results in enhanced expression of Cav-1 in both CD4+ and CD8+ T cells [68,69,70,83,98,99]. In complete lack of Cav-1 genes in CD8+, T cells cripple their function by LFA-1 redistribution and polarity [69,70]. Based on these findings, higher Cav-1 would benefit the host by more active T cell function. However, hyperactivation of CD8+ T cells has not been reported in HIV-1 infected patients. The major caveat of such studies is complete lack of Cav-1. The function of most of the signaling molecules depends upon their relative concentration. A complete lack of a molecule suggests a prominent role in the system; however, it cannot provide a clear picture for its position. Future experiments should be carried out with a varied expression of Cav-1 to know its real function. Several molecules modulate the expression of Cav-1 by different mechanisms [86]. The expression of Cav-1 is activated or enhanced by microbes and microbial byproduct lipopolysaccharide (LPS) or oxidative stress [85,119,120,121], strengthening the notion that Cav-1 role in sensing environmental cues. Oxidized low-density lipoprotein (oxLDL) or simvastatin also upregulates the expression Cav-1 [75,122]. Furthermore, cholesterol is an important modulator of Cav-1 expression. In addition, cellular growth factors can modulate Cav-1 expression through transcriptional mechanisms [123,124,125,126]. Multiple studies suggest that both HIV infection and antiretroviral drugs induce reactive oxygen species (ROS) production in HIV infected individuals [121,127], which subsequently can result in induction of Cav-1 expression. Such regulation of Cav-1 expression implies that Cav-1 is a critical molecule for cell survival and adjustment to stress.

Oxidative Stress: Oxidative damage has been linked to cellular senescence in aging animals as well as stress induced premature senescence. Cellular senescence is a mechanism of irreversible growth arrest to protect against proliferating aging cells and/or damaged cells to halt transmission of damage to daughter cells [128]. The HIV proteins Tat, Nef, Vpr, and gp120 have been shown to independently increase ROS while decreasing antioxidants establishing HIV-mediated oxidative stress [Reviewed in [127]]. Extensive studies also suggest that cART is a major contributor of ROS increase in HIV infected patients [127,129,130,131,132,133,134]. ROS is involved in a variety of cellular processes including proliferation, differentiation, host-defense, and wound healing ((Reviewed in [135]). The levels of increased ROS production is regulated by antioxidants such as superoxide dismutase glutathione peroxidase, catalase and vitamins E and C, and glutathione [135]. Uncontrolled increased levels of ROS lead to damage of macromolecules and causes cellular apoptosis and senescence [136,137,138,139,140,141,142,143,144]. ROS, therefore, promotes oxidative stress, which then results in cellular dysfunction and tissue destruction. Enhanced ROS production by HIV, thus, will advance the breakdown of cellular tight junctions of the epithelium at the mucosal surface as well as the blood-brain barrier vasculature, amplifying HIV infection. Furthermore, increased levels of ROS production due to HIV infection and/or cART treatment contribute to HIV associated vascular disease. Therefore, oxidative stress is a central contributing priming factor to many parameters leading to pathogenesis in HIV infected individuals irrespective of ART treatment. Oxidative damage has been linked to cellular senescence in aging animals as well as stress induced premature senescence.

Cav-1 and HIV inhibition: Caveolin 1 (Cav-1) plays a major role in controlling cellular senescence [119]. Several studies have shown that oxidative stress upregulates Cav-1 and enhanced expression of Cav-1 plays a central role in stress-induced cellular senescence [119]. Therefore, in response to environmental cues, Cav-1 can be induced to promote growth arrest of cells. Relevant to such notion, low expression of Cav-1 expression correlates with transformed cells, and inverse relationship between Cav-1 expression and cell transformation is established [83,145,146,147]. Furthermore, Cav-1 modulates cell cycle progression [72,83,145,148,149,150]. Cell cycle analyses show that Cav-1 maintains cell cycle at the quiescent stage, whereas the deletion of Cav-1 results cell cycle to progress from G0 to G1 and G2/M phases [72,83,148]. For example, Caveolin 1 is important in the regulation of cyclin D [149,151]. Previous studies of gene expression patterns in combination with epigenetic information revealed that the cell cycle regulator cyclin D2 might have an important role in maintaining the HIV latency [152], implicating a role for Cav-1 plays in HIV latency. Similar to other activators, we and others have demonstrated that HIV infection induces Cav-1 expression in macrophages [153]. Since the recent finding that Cav-1 expresses in human lymphocytes, we have also established HIV infection enhances the expression of Cav-1 in T cells (Unpublished results). Furthermore, we have previously demonstrated inhibition of HIV replication in Cav-1 over-expressing primary CD4+T cells and monocyte-derived macrophages [115,153,154]. The mechanism of induction of Cav-1 during HIV infection is not completely understood. Lin et al. [153] established that Tat is important in Cav-1′s upregulation at the transcriptional level, and this upregulation involves p53 protein. The study further shows that the p53 expression level is not affected by HIV infection. However, the phosphorylation of p53 at Ser15 and Ser46 is enhanced significantly. This suggests that the level of p53 activity plays an important role enhancement of Tat-mediated Cav-1 expression. Inhibitor of the p38 mitogen-activated protein kinase (MAPK) blocked the phosphorylation of the p53, subsequently reducing the induction Cav-1 significantly. These results suggest that in HIV-infected cells, Tat mediates activation of p38 MAPK, promoting the phosphorylation of p53, subsequently upregulating Cav-1 expression. However, further studies are needed to determine the factors and pathways involved in the mechanisms of Cav-1 up-regulation by HIV. These include epigenetic elements such as upstream factors as well as transcription factors and cis-acting elements. For example, the promoter region of Cav-1 NF-KB cis-element important for Cav-1 expression and regulation has not been examined in the context of the induction of Cav-1 expression mediated by HIV infection.

Caveolin-1 present on the cells surface and acts as an entry point for various infectious agents, including HIV-1. HIV-1 infects immune cells, mainly dendritic cells, macrophages, and T cells. The virus spread either by actively transferred to T cells from dendritic cells [155], direct interaction of virion with their cognate receptor, or through the virological synapses (VS) [156]. Dendritic captured virion presented on the dendrites, which are formed with actin filaments, and inhibition of Caveolin-interacting tetraspanin and dynamin prevented the process [155]. When a free virion interacts with a CD4/CXCR4 on T cells, they form a cluster; a requisite event for the viral attachment is controlled by Filamin-A, a Caveolin-interacting molecule [157,158]. These interactions among viruses and other cytoskeletal molecules are critical for transporting the cytoplasm for replication and finally reaching the nucleus. There is a defined role for the T cell synapse to occur for the caveolin-1 [70], which is needed for T cell activation. It is still unclear whether virological synapse on T cells for viral spread requires Caveloin-1; however, for efficient VS to form, it requires various actin-associated proteins that may be interacting with the Cav-1. Like the entry, viral budding also needs actin cytoskeletal rearrangements attached to Caveolin-1 at the surface [159]. Similarly, there is limited information on the mechanism of inhibition of HIV replication by Cav-1. Since Cav-1 participates in many cellular functions, the inhibition of HIV can involve several mechanisms (Figure 2). Two independent studies have demonstrated that Cav-1 inhibits HIV replication by transcriptional repression mediated through NF-κB [115,160]. However, the upstream factors involved in transcriptional repression mediated through NF-κB by Cav-1 is not known. Wang et al. [161] has proposed a potential mechanism for Cav-1’s ability to inhibit HIV replication that involves the association of Cav-1 and the HIV envelope. Cav-1 significantly suppressed Env-induced membrane hemifusion by possible interaction with the gp41. The env glycoprotein stimulates viral transcription and increases infection by modulating cellular machinery [162]. The increase in Cav-1 expression during HIV infection along its possible interaction and sequestration of Env can result in inhibition of Env mediated manipulation of cellular machinery to stimulate viral transcription and infectivity of virus progeny, subsequently contributing to HIV latent infection. Since surface lipid composition is essential in cell fusion of the block Env mediated fusion with target cells during an increased expression of Cav-1 may have to do with Cav-1’s role in cholesterol metabolism. Cav-1 is an essential regulator of cholesterol metabolism. Furthermore, Cav-1 is vital in cholesterol transport from the ER to the cell membrane. An increase in Cav-1 expression leads to the restoration of cholesterol efflux impaired by HIV via Nef and negatively affects virus replication [154]. Therefore, the specific interaction of Env with Cav-1 blocking fusion with target cells and the role of Cav-1 in cholesterol trafficking have important implication the establishment of HIV persistent in the patient under cART. Since cholesterol plays an important role in the resting or activation TCR, the HIV Env binding and/or fusion can lead to similar signal transduction affecting cell physiology. Furthermore, since Cav-1 is engaged in cholesterol efflux, an increase in its expression can modulate the confirmation stage of the lipid rafts and essential component of the lipid raft platform for the recruitment of signaling proteins to the plasma membrane. These changes would have an impact on several cellular functions, including the resting and cycling status of cells and including the establishment of HIV latent infection (Figure 2).

## 5. Conclusions

Cav-1/NCCLR membrane compartmentalization, along with its interaction with the cytoskeleton, can be an important environmental cue sensing mechanism, including ligand binding and HIV infection. The system regulates the steady and activation states of a cell equilibrium. The induction of Cav-1 expression can be a cell surviving mechanism to push the equilibrium from activated to a steady state by promoting cells to move into the resting stage. During HIV infection, most cells are overwhelmed by virus replication showing cytopathology, whereas a few cells become quiescent with latent infection. Since Cav-1 regulates cell cycle, signaling involving NCCLR can be important and play an essential role in promoting infected cells to become quiescent, subsequently resulting in latent infection as a cell survival mechanism (Figure 2). The upregulation of Cav-1 during viral infection can alter cholesterol composition modulate NCCLR, serving as initial danger signal and consequently leading to cell arrest and repression of viral transcription contributing to latent infection.

## Figures and Tables

**Figure 1 pathogens-09-00896-f001:**
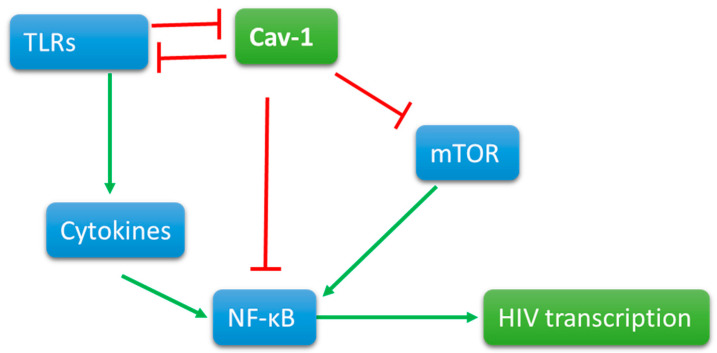
Cav-1 is present on the cell surface interactions with signaling molecules. When triggered, signaling molecules, such as Toll-like receptors, activate directly and indirectly by cytokine production NFkB. Viral promoter at the 5’ end of the proviral DNA contains the promoter that has NFKB binding site for viral transcription. Cav-1 directly or indirectly interacts with TLR signaling and mTOR pathways to reduce the activation of NFkB and thus the transcription of proviral DNA.

**Figure 2 pathogens-09-00896-f002:**
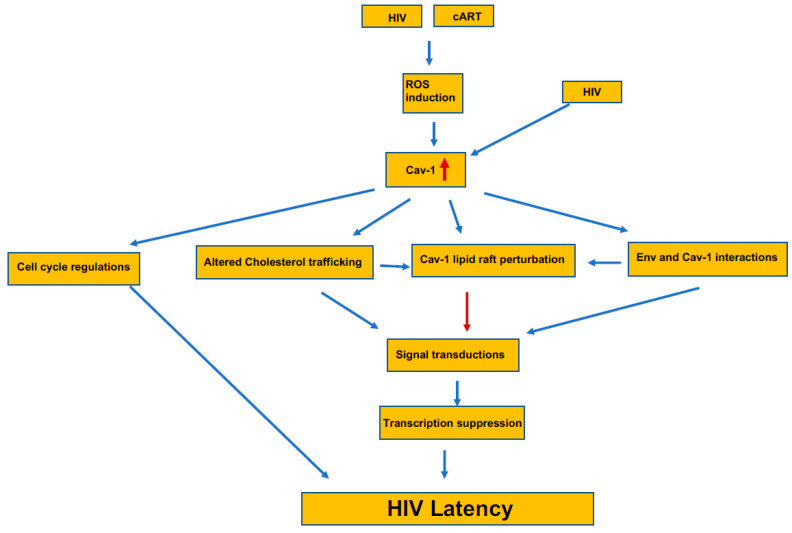
The induction and the pathway for the potential contribution of Cav-1 to the establishment of HIV latency. The presence of Caveolin-1 increases in HIV-1 infected and in latently infected cells by the generation of ROS. This increase in Cav-1 interferes with the cell cycle regulation that adds to the HIV latency. Simultaneously, increased Cav-1 alters cholesterol trafficking and thus the lipid raft composition, which directly and indirectly interferes with the interaction with Cav-1 and the virus’ envelop. Changes at the lipid raft that recruits signaling components alters the overall cellular signaling and initiates transcriptional suppression that further adds to the latency process.

**Table 1 pathogens-09-00896-t001:** Functions of Cav-1.

Functions of Cav-1	Reference
Component of lipid rafts	[71]
Cell cycle regulation	[72]
Cell signaling	[79,114,115]
Endocytosis	[76]
Cholesterol trafficking	[73,74,78]
Cytoskeletal rearrangement	[113]
Mechanosensing	[81,82]
B cell signaling	[83]
T cell signaling	[83]
Cytokine production	[85,114]
Cellular senescence	[72]
Cell death	[74]

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
