# Peer review of "The Potential Contribution of Caveolin 1 to HIV Latent Infection"

_pathogens, 2020, doi:10.3390/pathogens9110896_

Round 1

Reviewer 1 Report

The authors provide a comprehensive review of Calveolin-1 in HIV latent infection.

The manuscript would be improved by incorporating the following points:

  1. In section 1. Introduction the block and lock description should be expanded from one sentence to match the shock and kill description.  A recent review of HIV cure block and lock therapeutics provides a current description of all the block and lock therapeutics under investigation. See doi: 10.3389/fcimb.2020.00424.
  2. Page 2, line 87; Please fix the typo. "implications I HIV latent infection".
  3. Please consider including another figure to further describe the literature on Calveolin-1 and NF-kB in the HIV-1 LTR.
  4. Also consider expanding the section on the Calveolin-1 and actin cytoskeleton interaction, as HIV-1 virus entry is also reliant on the actin cytoskeleton. 

Author Response

We are asked to resubmit our manuscript with minor revision.  We have addressed all the minor issues raised by the reviewers as follows:

Reviewer 1

  1. In section 1. Introduction the block and lock description should be expanded from one sentence to match the shock and kill description.  A recent review of HIV cure block and lock therapeutics provides a current description of all the block and lock therapeutics under investigation. See doi: 10.3389/fcimb.2020.00424.

Response:  as suggested we expanded the section

  1. Page 2, line 87; Please fix the typo. "implications I HIV latent infection

Response: We fixed the typo as suggested.

  1. Please consider including another figure to further describe the literature on Calveolin-1 and NF-kB in the HIV-1 LTR

Response: We included a Figure (Figure 1) as suggested by the Reviewer.

  1. Also consider expanding the section on the Calveolin-1 and actin cytoskeleton interaction, as HIV-1 virus entry is also reliant on the actin cytoskeleton

Response: As suggested we expanded the Calveolin-1 and actin cytoskeleton interaction, as HIV-1 virus entry is also reliant on the actin cytoskeleton

Reviewer 2 Report

Bikash Sahay and Ayalew Mergia reviewed the literature on the role of Caveolin-1 in HIV latent infection. The review is interesting and extensive. At some points I found the review difficult to follow and therefore have suggestions for improving the readability of the paper.

  1. Please add sub-headings. For instance, in section 4 there is a paragraph on oxidative stress. Please add a sub-heading entitled oxidate stress.
  2. Please add a Table or Figure on the function of caveolin-1.
  3. The caption in Figure 1 is only a single line. Please extend and describe all processes and an explanation in the caption. Please explain the meaning of ROS.

Author Response

Reviewer 2

  1. Please add sub-headings. For instance, in section 4 there is a paragraph on oxidative stress. Please add a sub-heading entitled oxidate stress.

Response: We added sub-headings as suggested by the Reviewer.

  1. Please add a Table or Figure on the function of caveolin-1.

Response: We added a table as suggested by the Reviewer

  1. The caption in Figure 1 is only a single line. Please extend and describe all processes and an explanation in the caption. Please explain the meaning of ROS.

Response:  We have extended and described all processes and an explanation in the caption as suggested by the Reviewer.  We have also explained the meaning of ROS.